# Contribution of Geosciences to Cultural Heritage Conservation Assessment: The Case Study of the Loggiato dei Cappuccini in Comacchio (Italy)

**Elena Marrocchino** [1,*], **Maria Grazia Paletta** [1], **Lorenzo Ferroni** [1], **Rino Manfrini** [2] **and Chiara Telloli** [3]

1. Department of Environmental and Prevention Sciences, University of Ferrara, C.so Ercole I d'Este 32, 44121 Ferrara, Italy; pltmgr@unife.it (M.G.P.); frrlnz@unife.it (L.F.)
2. Comune di Comacchio, SETTORE VI, Protezione Civile, Servizi alla Persona, Igiene e Decoro Dell'ambiente Urbano, P.zza Folegatti, 44022 Comacchio, Italy
3. Atomic Energy and Alternative Energy (ENEA), Italian National Agency for New Technologies, Energy and Sustainable Economic Development Fusion and Technology for Nuclear Safety and Security Department Nuclear Safety, Security and Sustainability Division, via Martiri di Monte Sole 4, 40129 Bologna, Italy; chiara.telloli@enea.it
* Correspondence: mrrlne@unife.it; Tel.: +39-3393807477

**Abstract:** Geoscience disciplines play a pivotal role in the assessment of the conservation state of Cultural Heritage to orient the subsequent restoration interventions. In this report, we exemplify the potential of petrographic and thermographic analyses for the evaluation of the conservation state of a unique symbol of the architectural heritage in the challenging lagoon environment of Comacchio city (Ferrara Province, northeastern Italy). This study focuses on the Loggiato dei Cappuccini, starting from the historical analysis of the maintenance and restorations that this simple and pleasant monument has undergone over time. The degradation morphologies and the related triggering causes, characterized by macroscopic observations, were contextualized based on the recent restoration interventions. The current state of conservation has been evaluated quali-quantitatively, combining optical petrographic analyses of the main construction materials with thermographic analyses of the structures. The results of this study highlight the detrimental effects of previous restoration interventions on the long-term conservation state of the monument, emphasizing the need for a general environmental evaluation preliminarily to any conservative action. In particular, geoscience can contribute to a knowledge-based choice of materials that minimize the risk for alveolization and detachments.

**Keywords:** petrography; Comacchio (Italy); thermography; degradation phenomena; alveolization





## 1. Introduction

Cultural heritage derives from the development and diversification of human cultures and abilities and plays an important role in contemporary societies as a memory of their past and as a way of keeping their individual identity. In fact, the contribution and cross-cutting nature of cultural heritage for achieving a smart, sustainable, and inclusive territorial development is increasingly being recognized by policy [1–3]. Furthermore, it is also widely acknowledged that cultural heritage is an irreplaceable component of the socio-cultural and economic capital of a country, invaluable for the cohesion of communities, as well as for the creation and enhancement of social capital, economic impact, and environmental sustainability [4–9].

The importance of cultural assets for tourism and science, therefore, play a fundamental role in boosting the economy and knowledge development of a country: it can provide income by serving as a tourist destination, but also requires funds for conservation. Both economic factors are good reasons to keep cultural heritage objects in good condition.

Furthermore, as suggested by several authors [10–14], part of geo-conservation lies at the interface with cultural heritage, as the introduction of certain conservation measures can call for a good understanding of the local cultural environment, including the intangible heritage of indigenous societies, while the preservation of the built cultural heritage calls for appropriate consideration of the built heritage.

However, the vulnerability of cultural heritage has increased over time, and its exposure to a range of slow- and sudden-onset natural and human-induced hazards threatens its existence. Historic buildings, like all materials exposed to the external environment, are interconnected with surrounding abiotic and biotic factors, which include the pollution and damage caused by humans. Over time, all stone materials succumb to spontaneous decay due to their intrinsic characteristics, such as the mineralogical composition of the rocks, and the extrinsic effects of climatic agents [15–17]. Subject to environmental conditions, stone materials undergo physical and chemical degradations occurring in the form of deterioration phenomena, such as cracks, physical disintegration, abrasion, and detachment, which may also promote biological colonization.

Scientific investigation of cultural heritage has several important aims: (1) increasing our knowledge and understanding of the tangible heritage; (2) assembling information, including physical evidence, in order to add cultural value to the heritage, including the "intangible" values; and (3) helping to define the conditions, limits, risks, and potential for sustainable conservation and management of human heritage in general.

This report aims to exemplify the potential of petrographic and thermographic analyses for the assessment of the conservation state of a unique symbol of the architectural heritage in the challenging lagoon environment of Comacchio City (Ferrara Province, northeastern Italy). The eastern part of the territory of Ferrara province can generally be divided into two parts: one consisting of cultivated land, and the other of the residual fishing valleys of Comacchio and Volano. Comacchio is a small city laying on the south side of the present delta of the Po River. It is an early Medieval settlement, well-known because of the presence in its territory of several Etruscan cemeteries [18]. Comacchio has been shaped by a combination of factors including changes in sea levels and the course of the Po River, subsidence phenomena, and human activity over the past 5000 years [19–22].

In this context and starting from the historical analysis of the maintenance and restorations, this study focuses on the decay phenomena of the Loggiato dei Cappuccini, a simple and pleasant monument that is a symbol for the city of Comacchio (Figure 1). Macroscopic observations helped characterize the degradation morphologies and the related triggering causes, which were contextualized based on the recent restoration interventions. Quali-quantitative analyses, combining optical petrographic analyses of the main construction materials with thermographic analyses of the structures, were used to define its current state of conservation. The results highlighted the detrimental effects of previous restoration interventions on the long-term conservation state of the monument, emphasizing the need for a general environmental evaluation preliminarily to any conservative action. In particular, the contribution of geoscience is exemplified for the purpose of knowledge-based choice of materials that minimize the risk for alveolization and detachments. Geoscience disciplines can give helpful information to hypothesize suitable remedies for effective intervention projects.

## 2. Materials and Methods

### 2.1. Study Site and Visual Inspection

The city of Comacchio is located in the eastern part of the province of Ferrara (Emilia Romagna region in the northeast of Italy); its entire municipal territory area corresponds to a sector of the ancient Po Delta.

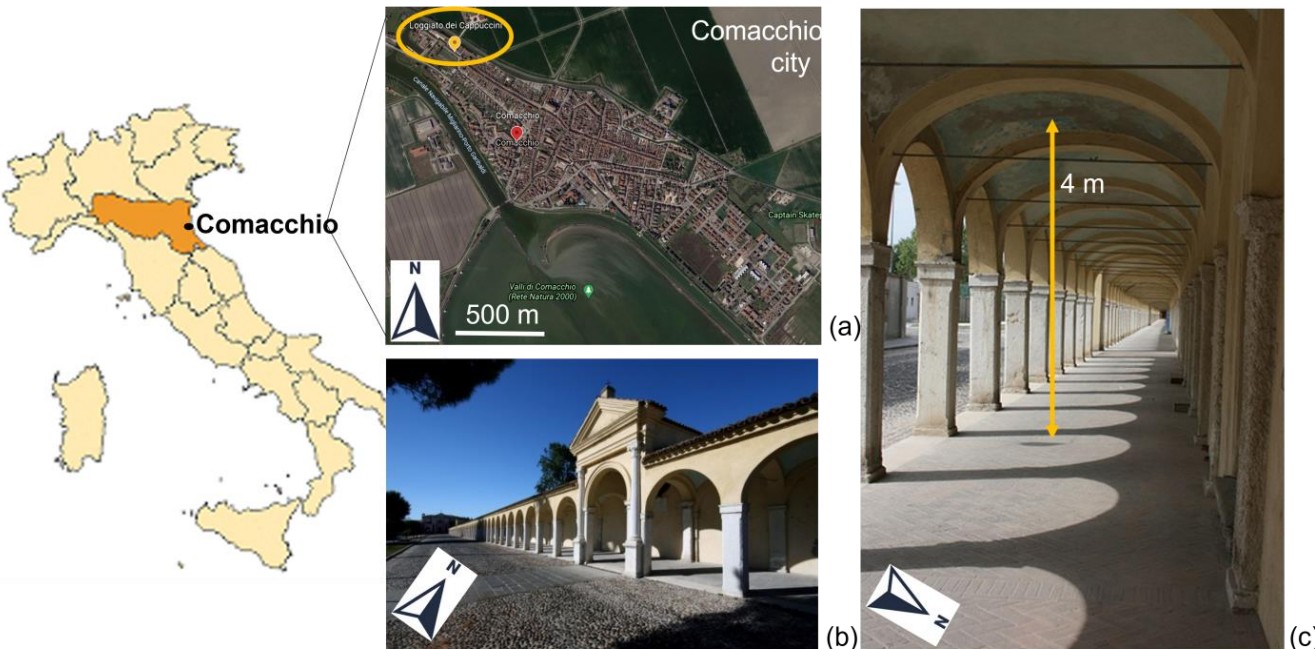

**Figure 1.** Map of Italy and the study site. Colored in orange is the Emilia Romagna region. (**a**) Comacchio aerial map (from Google Earth) and circled in yellow the Loggiato dei Cappuccini location; (**b**) detail of the front part of the Loggiato dei Cappuccini; (**c**) internal detail of the Loggiato dei Cappuccini.

This study site was the Loggiato dei Cappuccini, which was built, with both civil and military defense intentions, between 1646 and 1647 and consists of a succession of 142 arches [23,24]. Figure 2 shows a series of photographic illustrations of the Loggiato dei Cappuccini: Figure 2a,c,e,g, before the 2009 restorations, obtained from the municipality of Comacchio, and Figure 2b,d,f,h, the analysis campaign in 2019.

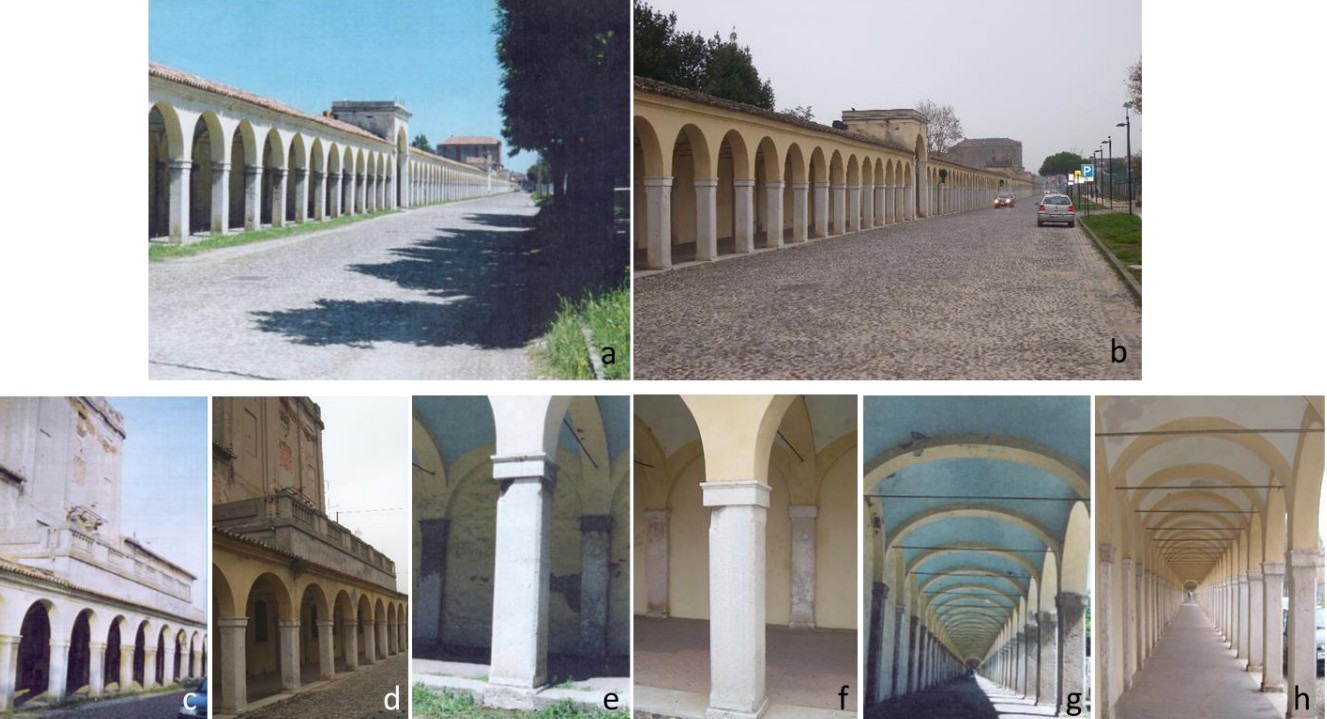

**Figure 2.** Pictures of different part of the Loggiato dei Cappuccini: (**a,c,e,g**), before the 2009 restorations (images from the municipality of Comacchio); (**b,d,f,h**), the corresponding current state condition.

The analysis campaign for this study is based on the preliminary inspection carried out in the summer/autumn 2015, the date on which degradation phenomena present in the structure were observed and for which it was decided to start the procedures to determine the causes of these phenomena of degradation.

As shown in Figure 3, the monument in 2019 was affected by the following forms of decay:

- lack of plaster placed on the internal wall of the Loggiato (Figure 3a,b);
- degradation on the internal columns: formation of black crusts (Figure 3c), alveolization (Figure 3d), physical disintegration (Figure 3e,f), and flaking (Figure 3g);
- degradation of the painting layer (Figure 3h,i);
- degradation caused by trees not being cut back (Figure 3j).

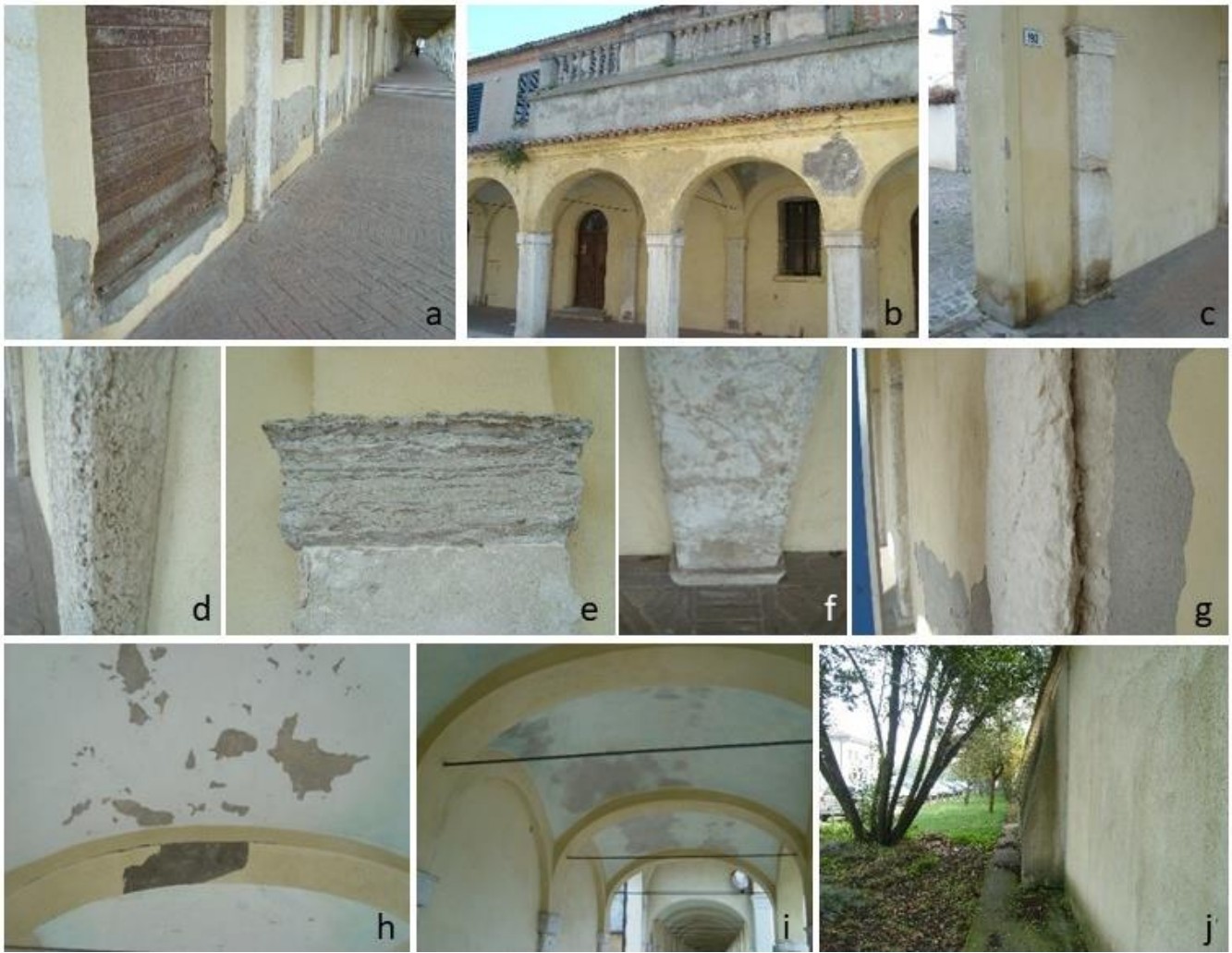

**Figure 3.** Degradation phenomena observed on the Loggiato dei Cappuccini during the analysis campaign on summer/autumn 2015: (**a**) degradation of the plaster placed on the internal wall; (**b**) lack of material on the external wall; (**c**) black crust at the base of the column; (**d**) alveolization phenomenon on the columns; (**e**) physical disintegration on the top of the columns; (**f**) physical disintegration on the basement of the columns; (**g**) flaking of the columns along the sedimentation floors; (**h**,**i**) degradation of the painting layer; (**j**) degradation caused by trees.

After this campaign, restorations were carried out relating to the replacement of the plaster of the internal wall and to the cleaning of the black crust on the columns.

Based on the visual analysis campaign in 2015 and the subsequent restoration, in September 2019 a further analysis campaign was carried out on selected parts of the

Loggiato, with emphasis on the deterioration of plasters, formation of crusts and patinas on the internal marble columns, and state of the painted layer of the vaults.

During the sampling period, the average temperature was 20.3 °C and the relative humidity was 78.4%; the data were obtained by Regional Environmental Protection Agency (Agenzia Prevenzione Ambiente Energia Emilia Romagna—ARPAE) stations. In the same way, the ARPAE stations detected that PM10 was about 20 µg m$^{-3}$ and PM2.5 was about 11 µg m$^{-3}$. The samplings had a duration of around a couple of months, between March and April (https://www.arpae.it/it/dati-e-report/report-ambientali/annuari-dellemilia-romagna/dati-ambientali-2019-la-qualita-dellambiente-in-emilia-romagna/view, accessed on 10 December 2022)

Figure 4 shows the different kinds of samples collected at the main entrance of the Loggiato dei Cappuccini, and Table 1 reports a detailed description of the type of samples and the corresponding analytical techniques. The samples collected were just small pieces of loose material scratched off the wall. Based on the visual analysis campaign, the collected samples were representative of the entire monument.

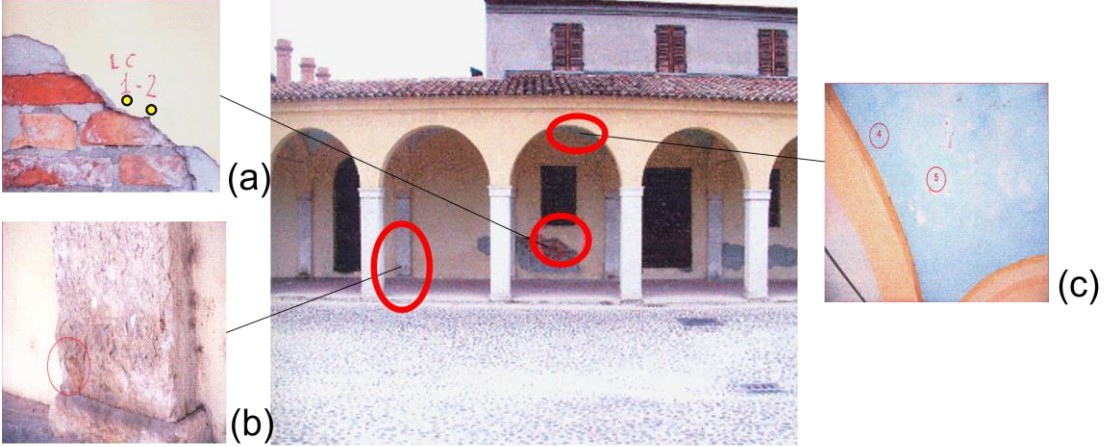

**Figure 4.** Sampling spots on the wall of the Loggiato dei Cappuccini: (**a**) samples LC-1 and LC-2 collected on the internal wall; (**b**) sample LC-6 collected on the basement of the column; (**c**) samples LC-4 and LC-5 collected on the vault. In the yellow circles the specific area where mercury intrusion porosimetry analysis was done.

**Table 1.** Detailed description of the samples collected on the internal part of the Loggiato, location, material investigated, type of decay, and techniques applied. Thermogravimetry and Differential Thermal Analysis (TG-DTA); Optical Transmitted Light Polarized Micro-scope (OTLPM).

| Sample ID | Location of the Sample Collected | Material Investigated | Type of Decay | Sample Preparation Method | Techniques |
|---|---|---|---|---|---|
| LC-1 | Wall | Plaster | Lack of material, moisture | Powdering for TG-DTA; thin section preparation for microscopic analyses | Conductivity; water content and dosage of soluble salts; mercury intrusion porosimetry; apparent density and total porosity; TG-DTA; OTLPM |
| LC-2 | Wall | Plaster | Lack of material, moisture | | |
| LC-4 | Vault | Pictorial layer | Degradation of the colors | Original sample | Stratigraphic analysis by stereomicroscope |
| LC-5 | Vault | Pictorial layer | Degradation of the colors | | |
| LC-6 | Basement of the column | Marble | Black crusts, alveolization, physical disintegration, flaking | Thin section preparation for microscopic analyses | OTLPM |

### 2.2. Surface Conductivity and Water Content Analyses

The distribution of surface conductivity was analyzed using a DELTA/OHM contact pycnometer mod. HD9213. The measurement was performed in the external layers of the masonry to better characterize the presence of moisture in the two survey sections, up to a height of two meters in the area shown in Figure 4, because the degree of decay is conceivably influenced by water content and amount of salt brought into solution.

The water content was further measured according to UNI EN 1015-17:2008 [25] and UNI EN 12274-2:2018 [26] to give more information related to the presence of moisture in addition to the previous analysis. The dosage of soluble salts was performed according to UNI EN 16581:2015 [27], UNI EN ISO 13788:2013 [28], and UNI 11088:2003 [29] to obtain data related to the presence of salt that could deteriorate the sampling site. Both the measurements were carried out on the masonry samples of the two sampling sections at different heights equidistant 0.5 m from the altitude or at the 2 m altitude in the area shown in Figure 4.

### 2.3. Porosimetric Analyses

A detailed analysis of porosity was performed on small pieces of plaster samples LC-1 and LC-2—Figure 4a. Mercury intrusion porosimetry analysis allowed the measurement of the volumetric distribution of open pores as a function of their diameter. The samples, dried and placed in a special container (dilatometer), were inserted into the porosimeter, where, after "emptying", mercury is introduced at an increasing pressure from 0.13 to 2000 Bar for permeation of the pores. The instrument records the volume of mercury introduced into the sample at various pressure values. The process also allows obtaining the average radius of the pores, the specific surface, and the porosity accessible to mercury. The determination of the apparent density (MVA) and of the total porosity was carried out according to UNI 11089:2003 [30] and UNI EN 12390-7:2021 [31].

### 2.4. Chemical and Petrographic Analyses

Chemical and petrographic analyses of the plaster (samples LC-1 and LC-2—Figure 4a), of the blue pictorial layer of the vault (samples LC-4 and LC-5—Figure 4b), and of the dark patina at the base of the column (sample LC-6—Figure 4c) were performed by means of thermogravimetric analysis (TG-DTA), differential scanning calorimetry (DSC), and petrographic analysis in thin sections, according to UNI 11530:2014 [32] and UNI 11305:2009 [33]. The thermogravimetric analysis measures the weight variations of the sample as the temperature rises in the range between 20 °C and 1000 °C and provides information on the thermal stability of the compounds that make up the sample in the same thermal range. Differential scanning calorimetry (DSC) measures the temperature variation of the samples analyzed with respect to a reference sample, providing information on the endothermic or isothermal reactions that occur in the same sample. The petrographic analysis on thin sections (20–30 μm thickness) allows to evaluate of the appearance of the binder, the lithological composition, and the granulometric distribution of the aggregate and the definition of some elements that have considerable importance in relation to the mechanical behavior of the materials, such as frequency and size of cavities and cracks, possible filling of fractures, and degree of degradation of materials.

### 2.5. Stratigraphic Analyses

Stratigraphic analyses of the blue painted layers (samples LC-4 and LC-5—Figure 4b) were performed with a polarized reflected light microscope, according to the UNI 11176:2006 [34] and UNI 11305:2009 [33] recommendation. In detail, observation of the surfaces of the samples was carried out using a stereomicroscope Optika SZ6745TR and an Optical Transmitted Light Polarized Microscope (OTLPM) Olympus BX51 on thin sections. The optical stereomicroscope (90× total magnification) equipped with MOTICAM 2500 5.0 M pixel webcam using the Motic Images Plus 2.0 ML software was used for reflected light observation on all the samples to define their structural aspect (grain size and texture), clasts dimensional,

and morphological aspect of their state of conservation [35]. In this specific work, OTLPM provided information about the quality of the work to produce the mortar [36,37].

## 3. Results

### 3.1. Surface Conductivity

The data obtained on the distribution of surface conductivity in the layers of the masonry of the internal wall of the Loggiato are shown in Figure 5. The iso-distribution of conductivity highlighted an anomalous area in correspondence with the gap in the plaster (colored in red in Figure 5 to be compared with Figure 4), which indicated that the finishing layers created a barrier to humidity (colored in green in Figure 5).

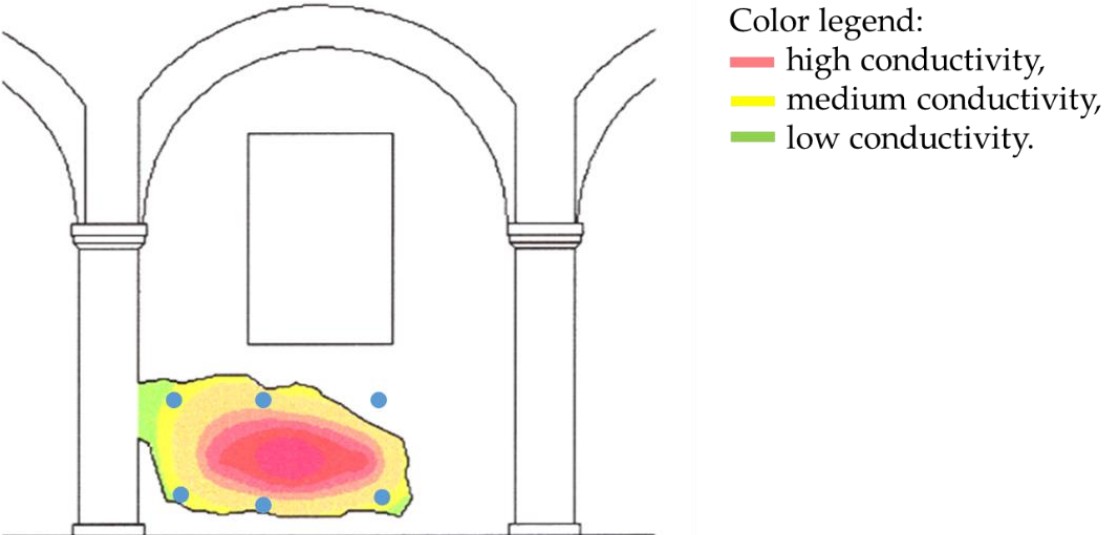

Color legend:
— high conductivity,
— medium conductivity,
— low conductivity.

**Figure 5.** Distribution of the surface conductivity in the part of the internal wall of the Loggiato selected for analysis. Compare the conductivity map with the state of the wall reported in Figure 4. Blue dots represent the sampling points; in the white area the instrument gave no measurements.

### 3.2. Water Content and Dosage of Soluble Salts

From the analysis of the internal wall of the Loggiato, the data relating to the water content and the dosage of soluble salts are shown in Table 2:

- Section 1 is the internal part colored in red in Figure 5, which is very humid (water content 3.29%), with anomalous values for the content of soluble salts (91.40 $\mu S\,cm^{-1}$).
- Section 2 is the external part colored in light yellow and green in Figure 5, which is wet (water content 5.88%), but with almost normal values for the content of soluble salts (60.35 $\mu S\,cm^{-1}$). For a clearer interpretation of the data obtained on the measurement of the water content and dosage of soluble salts (Table 2), it should be noted that generally dry brick masonry presents water contents of about 0.5–0.8%, values between 0.8–3% are typical of a humid context, values higher than 3% are considered typical of a very humid context, and values higher than 5% are due to a wet background. As for soluble salt contents, values lower than 50 $\mu S\,cm^{-1}$ indicate a very low presence of soluble salts, normal values are comprised between 50 and 70 $\mu S\,cm^{-1}$, and anomalous values are between 70 and 150 $\mu S\,cm^{-1}$. Higher values (>150 $\mu S\,cm^{-1}$) are treated as very anomalous.

**Table 2.** Mean and maximum value of the water content and the dosage of soluble salts relating to Section 1 (colored in red in Figure 5) and Section 2 (colored in light yellow and green in Figure 5).

|  | Water Content | | Soluble Salts | |
| --- | --- | --- | --- | --- |
|  | Mean Value (%) | Maximum Value (%) | Mean Value ($\mu$S cm$^{-1}$) | Maximum Value ($\mu$S cm$^{-1}$) |
| Section 1 | 3.29 | 5.59 | 91.40 | 139.6 |
| Section 2 | 5.88 | 13.78 | 60.35 | 92.2 |

The above values of water content and dosage of soluble salts are summarized in the iso-distribution maps shown in Figure 6a,b. Figure 6a shows the presence of moisture in the masonry: there is a greater presence of moisture in the lower right area (colored in brown and red), which rapidly subsides moving both leftward and upward. Regarding the map of the soluble salts (Figure 6b), in the lower part, they are found at medium-low values (blue and green colors) and increase moving upwards and toward the left part (orange color). The high salt content in the upper part is related due to the process of water evaporation, a well-known phenomenon that characterizes the rising moisture processes of saline solutions [38]. The comparative evaluation of the distribution of humidity and soluble salts showed the correlation between water content and concentration of soluble salts, enabling us to hypothesize that the dosage of the soluble salts should be inversely proportional to the humidity [39,40].

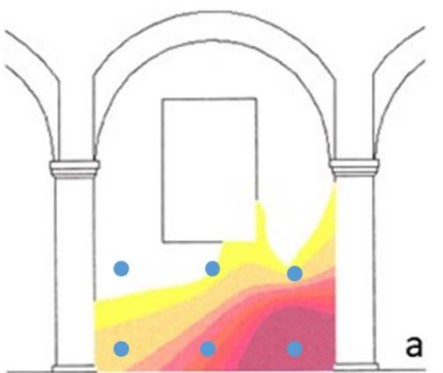 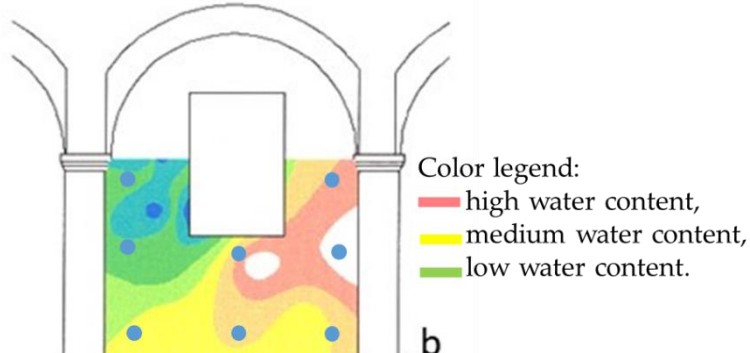

**Figure 6.** Thermograms on the analyzed internal wall of the Loggiato dei Cappuccini: (**a**) water content diagram; (**b**) soluble salts diagram. Blue dots represent the sampling points; in the white area the instrument gave no measurements.

*3.3. Plaster Porosity*

Mercury intrusion porosimetry analysis (Table 3), together with the determination of the apparent density (MVA) and total porosity on plaster (samples LC-1 and LC-2—yellow circle in Figure 4a) were also obtained. For the LC-1 sample, the values obtained are related to a mortar based on expanded resins [41], while for the LC-2 sample, a mortar was added with surfactants [42].

**Table 3.** Data of the cumulative volume (mm$^3$ g$^{-1}$), mercury intrusion porosimetry value (%), specific surface (mm$^2$ g$^{-1}$), average ratio ($\mu$m), apparent density (MVA—g cm$^{-3}$), and total porosity (%) of the internal wall plasters of the Loggiato analyzed (samples LC-1 and LC-2). N.d. not detected.

|  | Cumulative Volume (mm$^3$ g$^{-1}$) | Mercury Intrusion Porosimetry (%) | Specific Surface (mm$^2$ g$^{-1}$) | Average Ratio ($\mu$m) | MVA (g cm$^{-3}$) | Total Porosity (%) |
| --- | --- | --- | --- | --- | --- | --- |
| LC-1 | 97.48 | 9.7 | 5.43 | 0.854 | 1.46 | 40 |
| LC-2 | 93.35 | 9.3 | 5.52 | 0.895 | N.d. | 27.7 |

### 3.4. Chemical and Petrographic Features of Plasters

TG-DTA analysis provides information on the endothermic or exothermic nature of the reactions occurring in the sample. The comparison with an inert reference material provides semi-quantitative information on the chemical nature of the molecules involved in the thermal event. The analysis is carried out after drying the sample in an oven at 60 °C for 24 h and stabilizing it in the ambient atmosphere. The analysis of the plasters highlighted two different mixtures, which differed essentially in the percentage of dolomitic aggregates and organic substances (Table 4). Based on the comparison of the TG-DTA analysis results coupled with petrographic observations, it is possible to infer that the plaster samples presented different percentages of dolomitic compounds.

**Table 4.** TG-DTA results on the two plaster samples analyzed.

|      | Hydraulic Binder (%) | Dolomitic Aggregates (%) | Organic Substances (%) |
|------|----------------------|--------------------------|------------------------|
| LC-1 | 11                   | 62                       | 2.4                    |
| LC-2 | 11                   | 81                       | 1.6                    |

TG-DTA analysis provided information on the thermal behavior of samples, including the nature of the reactions occurring within them. By comparing the results of the analysis of plaster samples with the different percentages of dolomitic compounds, it is possible to identify differences in their thermal behavior that may be attributed to the presence of the dolomitic compounds [43].

The thin sections of plaster samples LC-1, LC-2, and LC-6 were observed by microscopic analysis. The petrographic observations are summarized in Table 5. LC-1 was a very coarse arenaceous granulometry mortar with color from hazelnut to brown and dark gray, with medium density. The distribution of aggregates seems rather even and regular across the field of view (Figure 7a,b). The aggregate, with medium sphericity, had a carbonatic composition; fragments of limestone and dolomitic limestone from micritic to microsparitic were recognizable. The groundmass shows a hydraulic appearance, with a colloform texture and an inhomogeneous structure. The adhesion between the matrix and the aggregate was good. Altered granules of unreacted cement with a diameter of 60–100 μm were observed. The porosity was quite high, deriving from the presence of the expanded resin, as well as from spherical vacuoles, typically due to the use of aerating additives, and microcracks were also observed. The storage conditions were found to be fair, even if the grade of weathering was identified as strong. The sample can be classified as premixed cement-lightened mortar.

**Table 5.** Detailed characteristics obtained by petrographic analysis on the collected samples.

| Sample Name | Texture Type     | Short Definition                     | Grain Size Class | Type of Alteration | Grade of Weathering Identified | Microcracking |
|-------------|------------------|--------------------------------------|------------------|--------------------|--------------------------------|---------------|
| LC-1        | granular         | premixed cement lightened mortar     | coarse           | chemical           | strong                         | perceptible   |
| LC-2        | granular         | premixed cement macroporous mortar   | coarse           | chemical           | strong                         | perceptible   |
| LC-6        | microcrystalline | biocalcilutite                       | fine             | chemical           | moderate                       | perceptible   |

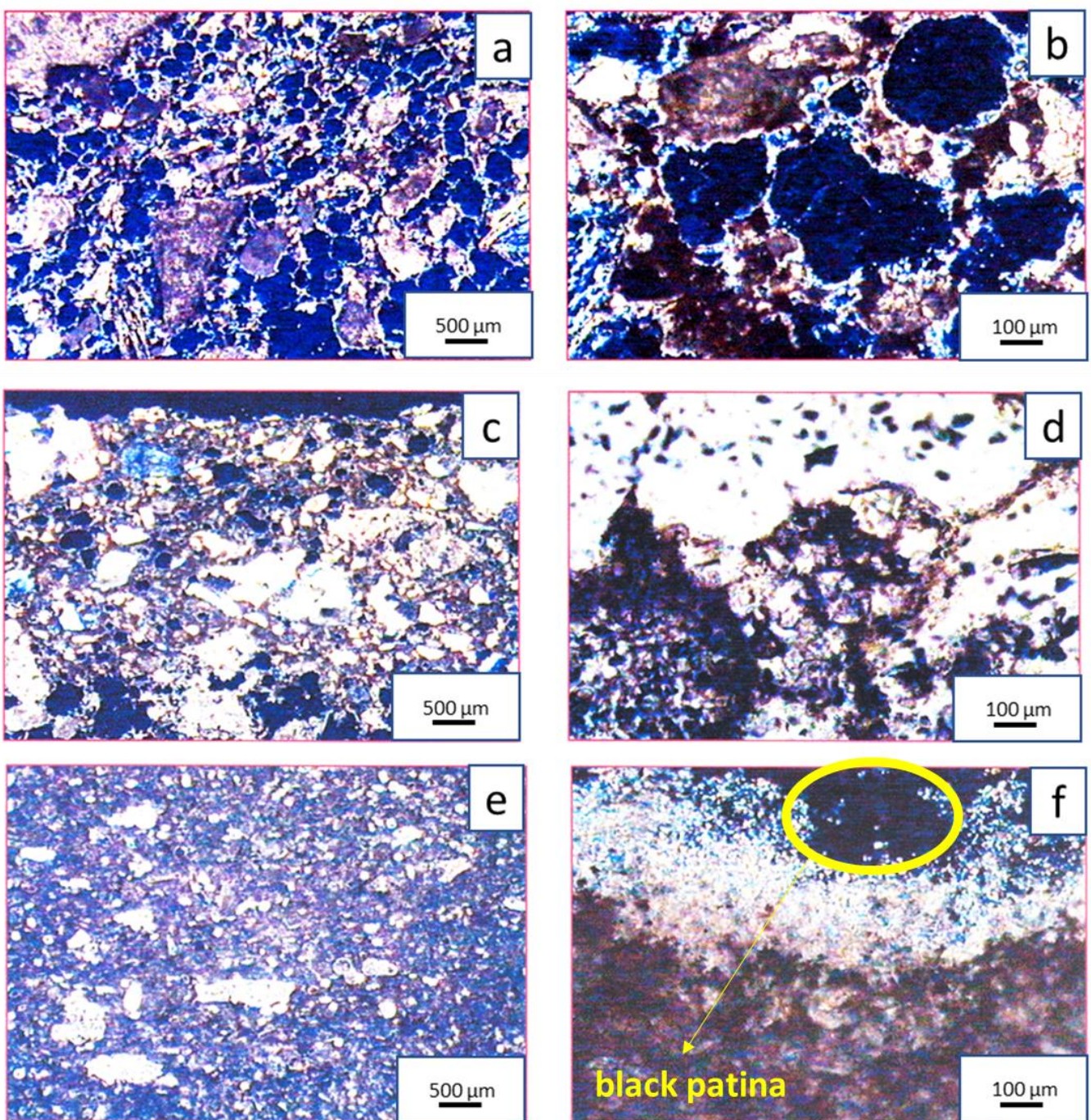

**Figure 7.** OTLPM characterization on the thin sections of the plaster samples of the internal wall of the Loggiato: (**a**) sample LC-1, magnification 70×; (**b**) sample LC-1, magnification 270×; (**c**) sample LC-2 magnification 110×; (**d**) sample PC-2, magnification 650×; (**e**) sample LC-6, magnification 70×; (**f**) sample LC-6, magnification 650×. Scale bar 100 μm colored in red in each figure.

LC-2 is a mortar with a coarse sandstone granulometry of hazelnut and gray color, with medium density and uneven distribution of the aggregate (Figure 7c,d). The aggregate, with medium sphericity, had an eminently carbonatic composition. Fragments of limestone and dolomitic limestone from micritic to microsparitic, and subordinate fragments of silicate rocks (especially gneiss) were recognized. There were some expanded resin granules. The groundmass shows a hydraulic appearance, with a micritic texture and an inhomogeneous structure. The contact between the matrix and the aggregate was good; it seems like gefuric

contact following the description in [44]. Altered granules of unreacted cement with a diameter of 60–100 µm were observed. The porosity was highly made from spherical vacuoles, deriving from aerating additives. The sample can be classified as premixed cement macroporous mortar with a strong grade of weathering.

LC-6 is micritic limestone rich in microfossils, including bryozoans, disarticulated valves of ostracoids, platelets of echinoderms, and rare foraminifera (Figure 7e,f). The incipient dolomitization essentially affected the larger bioclasts. A thin layer of recrystallized calcite was observed on the surface. The sample can be classified as biocalcilutite with a moderate grade of weathering.

Macroscopic observations revealed the presence of a black patina at the base of the columns, which can be related to dust and the deposition of organic matter [45,46]. Microscopic observation (Figure 7f) confirmed the presence of a black patina in one of the samples which seems to impregnate the whitish layer below. The whitish layer, micritic limestone, appeared to have increased porosity and decreased cohesion toward the patinated zone. On the basis of these observations, we can affirm that a localized chemical weathering of the natural stone, visible as a dissolution of the calcitic material where in contact with the black patina, has been detected. In the case of the Loggiato dei Cappuccini, the cause of the black patina can be associated with the rising dump in humidity from the ground and the alteration of biological patinas, or animal excrement.

The plaster samples of the Loggiato dei Cappuccini consisted of different layers: the rough coat (a lightened mortar) and the finish coat (a colored plaster). The rough coat is a very coarse sandstone mortar made up of carbonate aggregate from the crusher and cementitious binder. An expanded resin was probably added to the mixture—in a quantity sufficient to obtain the total porosity of 40%, as for other important Italian buildings [47]—and the results of the aerating additives were observed by petrographic analysis as spherical vacuoles, a common consequence from the use of these products.

### 3.5. Stratigraphic Features of the Painted Layer

Two different samples of the pictorial layer of the vault of the internal Loggiato dei Cappuccini were analyzed with respect to their stratigraphy.

Sample LC-4 (Figure 8a,b):

- LAYER A: overall white plaster, composed of binder and carbonate aggregate. The thickness was indefinable. The boundary with the overlying layer was blurred and the adhesion was good.
- LAYER B: pictorial layer of overall blue color. The thickness varied from 80 to 220 µm. No powdery blue pigments were clearly observed, from which the use of liquid organic pigments (probably phthalocyanine) was deduced. Rare red ocher grains were observed.

Sample LC-5 (Figure 8c,d):

- LAYER A: overall white plaster, composed of binder and carbonate aggregate. The thickness was indefinable. The limit with the overlying layer was instead clear and the adhesion was altered.
- LAYER B: pictorial layer of overall blue color, with evident superficial discoloration and patches of leopard in the mass. The thickness varied from 20 to 240 µm. No powdery blue pigments were observed, from which the use of liquid organic pigments (probably phthalocyanine) was deduced.

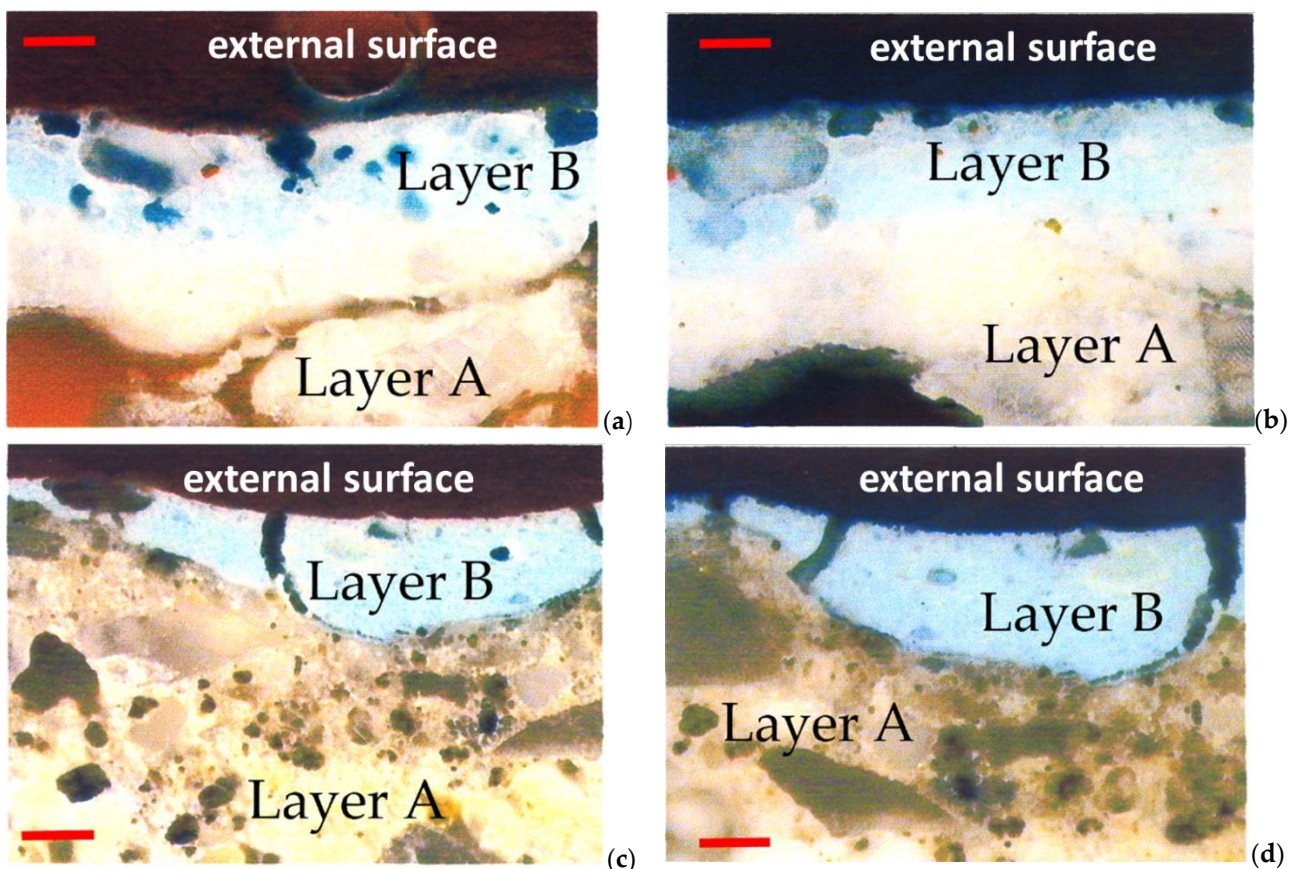

**Figure 8.** Stratigraphic analysis by stereomicroscopy of the pictorial layers: (**a**) sample LC-4 at 270× magnification; (**b**) sample LC-4 at 540× magnification; (**c**) sample LC-5 at 270× magnification; (**d**) sample LC-5 at 540× magnification. Scale bars 200 μm for images at 270× magnification and 100 μm for images at 540× magnification.

## 4. Discussion

In the context of the geoscience contribution to cultural heritage characterization [48], this work reports the results of the analysis campaign of the degradation phenomena on the Loggiato dei Capuccini, an important cultural heritage building of Comacchio. The research offers fundamental hints on the related triggering causes, in order to suggest appropriate remedies for future effective preservation and restoration actions. This research work was based on the observation of the degradation phenomena and related laboratory analyses, and the comparison with some of the main deterioration phenomena observed visually on the Loggiato as early as 2015. It emerged that the main forms of deterioration affecting the Loggiato concern the plaster (placed on the internal wall), the blue-painted layer of the vaults, and the detachment of the finishes.

Compared to the observations made in 2015, the deterioration of the plaster observed in the sampling of 2019 has expanded. In 2015, the phenomenon still appeared to affect only the finishing layer (Figure 3a), but a probable negative evolution in the underlying layers cannot obviously be excluded. Along the extension of the Loggiato, various traces of humidity were seen, located on the same internal masonry and on the surface of the plaster analyzed in our research work (Figure 4a).

Based on the petrographic characterization, the composition of plaster can play an important role in its susceptibility to degradation, such as (a) the presence of organic substances on the surface of the plaster may contribute to its degradation over time, in fact, substances can attract moisture and promote the growth of microorganisms that can damage the surface of the plaster [49]; (b) the use of aerating additives in the plaster can increase its porosity, which may make it more susceptible to weathering and deterioration over time.

The increased porosity can allow moisture to penetrate the surface of the plaster and cause it to expand and contract with changes in temperature and humidity [50]; (c) the chemical nature of the aggregates used in the plaster can also impact its durability [51]; (d) the presence of soluble salts in the plaster can also contribute to its degradation over time [52].

In addition, the porosimetry analyses allow to give few hypotheses that can be drawn about how weathering can impact on plaster porosity: (a) porous plaster may have a higher salt content in the pore water due to its increased ability to absorb and retain moisture from the surrounding environment. This can contribute to weathering by promoting the dissolution of the soluble salts in the plaster and accelerating their crystallization, which can cause damage to the plaster [53]; (b) the porosity of the plaster may also impact its ability to resist weathering by affecting the rate of water absorption and evaporation. Plaster with high porosity may absorb and retain more moisture, which can promote weathering due to the expansion and contraction of the plaster caused by changes in temperature and humidity [53]; (c) the distribution of soluble salts within the plaster may also be influenced by its porosity, with more porous plaster potentially having a higher concentration of salts in certain areas [53]. This can result in uneven weathering patterns and damage to the surface of the plaster. On the whole, the porosity of plaster plays an important role in its susceptibility to weathering. Further sampling and experimentation will be necessary to fully understand the mechanisms behind these processes and to develop effective strategies for preserving and protecting plaster surfaces.

The 2015 survey highlighted the phenomenon of blackening on the bases of the columns (Figure 3c), and the 2019 analysis campaign confirmed the persistence of the same phenomenon. Microscopic observation revealed the presence of a black patina. This appears to impregnate the underlying natural stone, reducing its cohesion. This may have important implications for conservation protocols; in fact, removing the patina will weaken the monument by exposing the chemically weathered, detached parts of the natural stone, thus accelerating its disintegration. For these reasons, based on this information, different restoration actions should be taken into account, for example: (a) leave the patina in place: given that removing the patina could accelerate the disintegration of the natural stone surface, one possible approach would be to leave the patina in place. This approach would involve taking steps to stabilize the monument and prevent further degradation, without attempting to remove the patina. (b) Test removal techniques: if removal of the patina is deemed necessary, a range of techniques could be tested to determine the most effective and least damaging method. For example, mechanical, chemical, or laser-based methods could be evaluated to determine which approach would be most appropriate for the specific type of stone and patina present. (c) Monitor the monument: regardless of whether the patina is removed or left in place, ongoing monitoring of the monument will be necessary to assess its condition and determine whether any additional conservation measures are required. Regular inspections could help identify any changes in the stone surface or patina and enable conservationists to take action before further damage occurs. (d) Assess the source of the patina: understanding the source of the patina could be important in determining the appropriate course of action. For example, if the patina is the result of pollution, steps could be taken to reduce emissions in the surrounding area to prevent further damage. Alternatively, if the patina is the result of a natural process, conservationists may need to take a different approach to address the issue [54,55]. In addition, the base of the columns showed signs of degradation, such as progressive flaking along the sedimentation planes, due to combined chemical-physical aggression phenomena—the latter related to wind activity, which is of particular importance in relation to the architectural conditions of the Loggiato. Indeed, images as in Figure 3c–f evidence that the phenomenon of alveolization is more present on the sides of the columns perpendicular to the inner masonry of the Loggia (Figure 3d). The circumstance suggests the formation of air currents inside the Loggiato, mixed with the components illustrated above, running through the closed side, which impact their load specifically and predominantly on the shorter sides of the pilasters. In the preservation actions of the Loggiato, it would be interesting and extremely useful to

be able to monitor the air quality. In any case, eliminating the triggers of degradation is certainly a tall order.

With regard to the presence of soluble salts in the walls analyzed, some assumptions can be made about the circulation of water: (a) the process of water evaporation contributes to the high salt content in the upper part of a solution [56]. This suggests that water circulates through evaporation and condensation processes; (b) the distribution of humidity and soluble salts is correlated, with higher humidity levels associated with lower concentrations of soluble salts [56]. This suggests that water circulation may be influenced by factors such as temperature and air pressure, which can affect the humidity levels in a given environment; (c) the dosage of soluble salts is inversely proportional to humidity levels, indicating that water circulation may be influenced by the concentration of dissolved salts in a solution [56]. This suggests that water may circulate through osmosis or other related processes, where water moves from areas of lower salt concentration to areas of higher salt concentration. Overall, these hypotheses suggest that water circulation is a complex process that is influenced by a variety of factors, including temperature, air pressure, and the concentration of dissolved salts in a solution. Further research and experimentation may be necessary to fully understand the mechanisms behind water circulation in various environments.

Relating to the pigment of the vaults, the investigations carried out concerned the presence of conspicuous discolored spots on the blue color. The analyses showed that the blue paint has a composition based on organic resin colored with a liquid organic pigment. This pigment could very likely belong to the Phthalocyanine family [57,58]. Further detailed analyses using SEM-EDS and μ-Raman will allow this aspect to be better defined, enabling more targeted and effective restoration work. The stratigraphy showed the complete superficial discoloration of the altered sample, to which a leopard spot discoloration is added that affects the entire pigmented thickness. In the "healthy" sample, the surface does not appear discolored, while some discolored spots can be observed inside. The investigations carried out suggest that the discoloration is most probably caused by the degradation of the organic chemical species [59,60] and depends on the stability of the pigment rather than being the consequence of a phenomenon of physical surface degradation [55]. The cause is therefore identified in the poor quality of the pigment used. The conservation of a historical artifact does not end with restoration work, since the phenomena of degradation are linked to the surrounding environment and the reactions that take place within it. In order to allow longer survival, continuous maintenance is indispensable to prevent the action of any degradation. In this context, it is important to develop an adequate communication of geological knowledge, which can significantly contribute to the construction of social knowledge for human communities, in order to extend this awareness to the entire civil society and promote the recognition of the usefulness of geosciences in everyday life [61–65]. In fact, according to Zafeiropoulos [61] and reference therein, by recognizing the value of a region's geological heritage, "geological culture and geoethics can strengthen the bonds between people and their land, between their places of origin and their own memories".

In regards to the Loggiato dei Cappuccini, analysis campaigns supported by geoscientific methods as proposed in this paper could be proposed to local authorities for planned periodic monitoring of those characteristics that have proved to be the main cause of the deterioration. Among these, it could be useful to check the physical-chemical characteristics of the plasters, the paint layers, and the stone of the columns, aimed at ascertaining the consistency of the restored surfaces and the presence of any soluble salts or efflorescence, or the presence of biodeteriogenic phenomena. The analysis campaign should also include observations and evaluations of static, aesthetic, and general decorum.

## 5. Conclusions

Geoheritage is an integral and inseparable part of natural heritage and has scientific, aesthetic, scenic, economic, and intrinsic values that must be preserved and passed on to future generations. Although in recent decades the regulation and public use of geoheritage have mainly taken place in protected natural areas, the geoheritage of urban areas also needs to be given due attention by research teams and protected by local administrations.

Urban geoheritage, defined as a set of unique geological elements, processes, and outcrops in and around cities, is important to modern society for its scientific, educational, and tourist value. In this light, the cultural heritage of the cities must also be considered part of the geoheritage of an area.

Geoethics is thus a tool for raising public awareness of issues concerning geopolitical resources and the geoenvironment. An ethical approach must emphasize the importance of nature as a sensual, contemplative, spiritual, religious, and aesthetic experience that is passed down to future generations, rather than just the economic viability of natural resources.

When studying cultural heritage, especially when addressing problems regarding complex materials, there is a necessity to undertake a shared, multidisciplinary approach. Within this framework, geosciences offer a wide range of competencies and specific knowledge of natural materials across a wide range of spatial and temporal scales [66–69]. Geosciences are fundamental to perceiving the fine interplay between human society and natural resources. The perception of the complexity of materials, the interaction between human activities and natural processes, and the availability and properties of geo-resources are important requisites for heritage studies.

In this light, the analysis campaigns on the building materials of the Loggiato dei Cappuccini in Comacchio permitted, starting from the historical analysis of the maintenance and restorations, to identify the degradation morphologies and the related triggering causes by macroscopic observations, to contextualize based on the recent restoration interventions. The results of this study highlighted the detrimental effects of previous restoration interventions on the long-term conservation state of the monument, emphasizing the need for a general environmental evaluation preliminarily to any conservative action.

In the case of the Loggiato dei Cappuccini, future analysis campaigns with the support of geo-scientific methods, such as those proposed in this work, could be proposed to the local authorities for planned periodic monitoring of those elements that have been shown to be the main cause of degradation.

In particular, in fact, scientific heritage research not only helps to increase our knowledge and understanding of tangible heritage but is also useful in raising public awareness of cultural resources and geoenvironmental issues in a given area.

**Author Contributions:** Conceptualization, E.M. and C.T.; methodology, E.M., C.T. and R.M.; validation, C.T.; formal analysis E.M. and R.M.; investigation, E.M., C.T. and R.M.; resources, E.M. and C.T.; data curation, E.M., C.T. and M.G.P.; writing—original draft preparation, E.M., C.T. and L.F.; writing—review and editing, E.M., C.T., L.F. and M.G.P.; visualization, E.M., C.T. and M.G.P.; supervision, E.M.; project administration, E.M. All authors have read and agreed to the published version of the manuscript.

**Funding:** This research received no external funding.

**Data Availability Statement:** Raw data that support the findings of this study are made available on request.

**Conflicts of Interest:** The authors declare no conflict of interest.

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
