# Peer review of "Contribution of Geosciences to Cultural Heritage Conservation Assessment: The Case Study of the Loggiato dei Cappuccini in Comacchio (Italy)"

_geosciences, doi:10.3390/geosciences13060157_

Round 1
Reviewer 1 Report
The paper deserves attention as an illustrative case of multidisciplinary approach based on geoscience disciplines applied to cultural heritage. Only few minor changes are requested.

Author Response
Dear Reviewer, we would like to thank you for your precious comments.
Here is our reply.
Best regards
*************************************************************************************Please spell out the words before to use acronyms.
- We thank the reviewer for the clarification, we added the definition before the acronym.
Temperature and humidity have been registered by the same stations? Also useful, add brief info about the duration of sampling period.
- We thank the reviewer, and we improved the text with some information as requested.
The words of these two acronyms could be spelled out in the caption.
- We thank the reviewer, and we improved the captions with the definition of the acronyms.
Line 165: Check if a verb is missing.
- We thank the reviewer, and we rewrote the sentence.
From 220 to 227; These periods should be moved to the beginning of 3.2
- We thank the reviewer, as suggested we moved the lines at the beginning of the paragraph.
Reviewer 2 Report
Title: Review report on "Geoscience Contribution to Cultural Heritage Conservation Assessment: the Case Study of the Loggiato dei Cappuccini in Comacchio (Italy)"
General Comments:
The paper titled "Geoscience Contribution to Cultural Heritage Conservation Assessment: the Case Study of the Loggiato dei Cappuccini in Comacchio (Italy)" presents a well-executed study that assesses the geoscientific contribution to cultural heritage conservation. Overall, the paper is well-written, and the methodology is proper. However, there are some minor revisions required to improve the paper's overall quality, including a more comprehensive literature review, including the discussion of geoethics and the tourism potential of the explored cultural heritage.
Specific Comments:
The authors need to provide a more comprehensive literature review to provide context to their study. They could include the references suggested in the review report to gain a better understanding of geoheritage as a cultural heritage. These references could be cited throughout the manuscript to strengthen the argument and demonstrate the significance of the study. I would recommend the following references:
· Pijet-MigoÅ„, E., & MigoÅ„, P. (2022). Geoheritage and cultural heritage—A review of recurrent and interlinked themes. Geosciences, 12(2), 98.
· Catana, M. M., & Brilha, J. B. (2020). The role of UNESCO global geoparks in promoting geosciences education for sustainability. Geoheritage, 12(1), 1.
· Melelli, L. (2019). “Perugia upside-down”: a multimedia exhibition in Umbria (Central Italy) for improving geoheritage and geotourism in urban areas. Resources, 8(3), 148.
· Vegas, J., & Diez-Herrero, A. (2021). An assessment method for urban geoheritage as a model for environmental awareness and geotourism (Segovia, Spain). Geoheritage, 13, 1-17.
· Marjanović, M., Tomić, N., Radivojević, A. R., & Marković, S. B. (2021). Assessing the geotourism potential of the Niš city area (Southeast Serbia). Geoheritage, 13, 1-18.
Additionally, the authors should include a conclusion section in the paper to summarize the findings and to provide the implications of the study for the cultural heritage conservation and geosciences field.
While the authors do not focus their paper on the (geo)tourism potential of the explored cultural heritage, it would be helpful if the authors could include some minor indicators in the discussion regarding the potential socio-economic affirmation of the explored cultural heritage for tourism as well as conservation. The authors should cite the relevant references in the discussion section to support their arguments.
Finally, the authors should include a section discussing geoethics as a potential factor that can impact the conservation of cultural heritage. The suggested references should be used to provide context and support for the discussion. I would recommend the following references:
· Zafeiropoulos, G., Drinia, H., Antonarakou, A., & Zouros, N. (2021). From geoheritage to geoeducation, geoethics and geotourism: A critical evaluation of the Greek region. Geosciences, 11(9), 381.
· Peppoloni, S., & Di Capua, G. (2012). Geoethics and geological culture: awareness, responsibility and challenges. Annals of Geophysics.
· Foresta Martin, F., & Peppoloni, S. (2017). Geoethics in science communication: the relationship between media and geoscientists. Annals of Geophysics.
· Bobrowsky, P., Cronin, V. S., Di Capua, G., Kieffer, S. W., & Peppoloni, S. (2017). The emerging field of geoethics. Scientific integrity and ethics in the geosciences, 175-212.
Overall, the paper presents an excellent contribution to the study of geoscience and cultural heritage conservation, and with the suggested revisions, the paper can be accepted for publication.
Author Response
Dear Reviewer, we are grateful for your precious comments.
Here is our reply.
Best Regards
********************************************************************************
Reviewer #2
General Comments. The paper titled "Geoscience Contribution to Cultural Heritage Conservation Assessment: the Case Study of the Loggiato dei Cappuccini in Comacchio (Italy)" presents a well-executed study that assesses the geoscientific contribution to cultural heritage conservation. Overall, the paper is well-written, and the methodology is proper. However, there are some minor revisions required to improve the paper's overall quality, including a more comprehensive literature review, including the discussion of geoethics and the tourism potential of the explored cultural heritage.
Specific Comments:
The authors need to provide a more comprehensive literature review to provide context to their study. They could include the references suggested in the review report to gain a better understanding of geoheritage as a cultural heritage. These references could be cited throughout the manuscript to strengthen the argument and demonstrate the significance of the study. I would recommend the following references:
- Pijet-MigoÅ„, E., & MigoÅ„, P. (2022). Geoheritage and cultural heritage—A review of recurrent and interlinked themes. Geosciences, 12(2), 98.
- Catana, M. M., & Brilha, J. B. (2020). The role of UNESCO global geoparks in promoting geosciences education for sustainability. Geoheritage, 12(1), 1.
- Melelli, L. (2019). “Perugia upside-down”: a multimedia exhibition in Umbria (Central Italy) for improving geoheritage and geotourism in urban areas. Resources, 8(3), 148.
- Vegas, J., & Diez-Herrero, A. (2021). An assessment method for urban geoheritage as a model for environmental awareness and geotourism (Segovia, Spain). Geoheritage, 13, 1-17.
- Marjanović, M., Tomić, N., Radivojević, A. R., & Marković, S. B. (2021). Assessing the geotourism potential of the Niš city area (Southeast Serbia). Geoheritage, 13, 1-18.
- We thank the reviewer for these fruitful suggestions, we implemented the manuscript with the indicated references.
Additionally, the authors should include a conclusion section in the paper to summarize the findings and to provide the implications of the study for the cultural heritage conservation and geosciences field.
- We agreed with the reviewer’s suggestion, and we split the “discussion and conclusion” paragraph into two distinct paragraphs
While the authors do not focus their paper on the (geo)tourism potential of the explored cultural heritage, it would be helpful if the authors could include some minor indicators in the discussion regarding the potential socio-economic affirmation of the explored cultural heritage for tourism as well as conservation. The authors should cite the relevant references in the discussion section to support their arguments.
Finally, the authors should include a section discussing geoethics as a potential factor that can impact the conservation of cultural heritage. The suggested references should be used to provide context and support for the discussion. I would recommend the following references:
- Zafeiropoulos, G., Drinia, H., Antonarakou, A., & Zouros, N. (2021). From geoheritage to geoeducation, geoethics and geotourism: A critical evaluation of the Greek region. Geosciences, 11(9), 381.
- Peppoloni, S., & Di Capua, G. (2012). Geoethics and geological culture: awareness, responsibility and challenges. Annals of Geophysics.
- Foresta Martin, F., & Peppoloni, S. (2017). Geoethics in science communication: the relationship between media and geoscientists. Annals of Geophysics.
- Bobrowsky, P., Cronin, V. S., Di Capua, G., Kieffer, S. W., & Peppoloni, S. (2017). The emerging field of geoethics. Scientific integrity and ethics in the geosciences, 175-212.
- We would like to thank the reviewer, for the valuable suggestions. We implemented the text of the manuscript, and we added the recommended references.
Overall, the paper presents an excellent contribution to the study of geoscience and cultural heritage conservation, and with the suggested revisions, the paper can be accepted for publication.
- We would like to thank you the reviewer for his/her worthwhile evaluation.
Reviewer 3 Report
MAJOR ISSUE
The paper needs much more detail description of the sampling procedure and of samples. For now, it is not clear whether the samples were just a loose material scratched off the wall, or monolithic blocks cut off from the wall? How large were the samples? How deep did the sampling penetrate into the wall? How were the samples further pretreated before the analyses – for example, cut into pieces before porosimetry, powdered before TG-TDA, thin sections prepared before microscopy, etc.? All these details must be provided.
Please also say more about your sampling strategy. For example, why two samples of plaster (LC-1 and LC-2) have been collected instead of one? What was the difference between both sampling spots LC-1 and LC-2visible in field? Did you try to track some particular difference, e.g. proximity of the damaged zone, intensity of damage etc.? And the same for other types of samples.
Another issue is that Discussion does only loosely refers to the presented results. Most of the results are not discussed in the Discussion, while them all should be. Please refer to your water content and salt content analyses, and basing on this please present hypotheses on the water circulation (or water and salt circulation) within the walls. Please also refer to the porosimetry analyses and say how the plaster porosity can impact the weathering. And please refer to the TG analyses and say how the composition of plaster may impact the degradation. Please also explain better your conclusion about the degradation of the pigment. Which exactly microscopic observations allow for such conclusion?
MINOR ISSUES
Page 1, lines 35-43 – it would be worthy to mention also the importance of cultural heritage for tourism and for science (to study the past). And that it has ambiguous economical importance: can provide income serving as a touristic destination, but also requires funds for conservation. Both these economic factors are good reasons to keep the cultural heritage objects in good condition.
Page 4, lines 107-108 – it is not clear how the 2015 campaign could reveal the 2019 damage, that is future in regard to 2015. Please rephrase these sentences.
Page 4, line 111 (and Figure 3 e,f caption, and anywhere further in the text) – the term “erosion” used here seems not to be the best choice. In general, erosion means the removal of material by erosional agent, such as water flow or wind. Here, it seems that you depict the effect of weathering (in situ deterioration of a rock), which looks at your photos like in situ physical disintegration. This would be a better term.
Page 4, line 130 – please note that Fig. 4 shows the sampling spots, not kinds of samples.
Page 5, lines 143-144 – please mark it precisely in Fig. 4 where the conductivity analysis was done.
Page 7, lines 214-217 – it is not clear if the information about the water content range in dry, humid etc. masonry is for the Loggiato samples only, or it is a general information about any bricks. If general, then citation(s) is needed. The same for salt content.
Page 7, lines 241-242 – “revealing that the dosage of the soluble salts was inversely proportional to the humidity” – could you please comment more on this observation in the Discussion section? In general, I would expect different correlations in different water regimes. So, if the water content distribution within the wall was constant over long time, then the zones of low water content are zones of evaporation. And within these zones the concentration of salt in water should be increased locally up to the saturation level. Alternatively, if the water content distribution within the wall was only a temporary situation, caused e.g. by a recent rainfall, then zones of low water content are those that water did not reach (yet) by its capillary action, so the amount of salts should be low due to a short time of dissolution. Can you please refer to such opinion, and also to the possible impact of weather on your observations?
Page 8, line 273 – “uneven distribution of the aggregate (Figs. 7a-b)” - looking at Figure 7 a,b, the distribution of aggregates seems rather even and regular across the field of view.
Page 8, lines 273-274 – “The crushed aggregate, with very angular medium sphericity” – there are several quite controversial statements in this sentence. First, what is crushed aggregate? If it was crushed, how do you know that it was originally an aggregate? Or if it is an aggregate right now, how do you know it was crushed? And also, what does it mean “angular sphericity”? Something is either angular or spherical. The same also in lines 284-285.
Page 8, line 276 - “bottom mass is hydraulic in nature” – you probably mean “ground mass” (or “groundmass”). And what does it mean a hydraulic nature? Do you mean the appearance or the parameters? The same also in line 288.
Page 9, line 277 – “The adhesion between the matrix and the aggregate” – if you describe the character of contact between the groundmass and larger clasts, please follow the micromorphological terminology provided by Stoops G. 2008. Micromorphology. In: Chesworth W. (Eds)., Encyclopedia of Soil Science. Springer, p. 457-466. The same also in lines 289-290. In your case of LC-1, it looks like gefuric contact (I am not sure about the LC-2 sample, the photograph is not the best).
Page 9 line 298 – a typo “bdeposition”
Page 9 line 298 – what is the argument behind the blackish staff to be an organic matter, and not the manganous oxide precipitations?
Page 10, line 314 – “The plaster samples of the Loggiato dei Cappuccini consisted of different layers” – please show microscope photos (or et least drawings) presenting the spatial relationships between variable layers and their thickness.
Page 10, line 339 – “No powdery blue pigments were observed” – aren’t the leopard spots the pigment powder grains? If not, please provide an explanation or hypothesis what can it be.
Figure 1 caption – “The Italian map” sounds weird, I would propose “The map of Italy” instead.
Figures 5 and 6 – why not to provide color scales for these figures?
Figure 6 – it is not clear how these thermograms were constructed. How many measurements exactly were used? Where exactly were these measurements done? This must be specified.
Figure 7 e,f – why the black patina is not shown at the photographs?
Figure 8 – please indicate where is the external surface.
Table 4 – how could you conclude on the basis of TG analyses that the dolomitic compounds were aggregates? Would other-than-aggregate form of dolomite within a sample produce different TG diagram?
Table 5 – the “Grade of weathering identified” is not explained in the text
English is Ok, but long sentences can be divided into shorter ones to improve the readability.
Author Response
Dear Reviewer, we would like to thank you for your precious comments and suggestions. Here is our reply.
Best Regards
*****************************************************************************
Reviewer #3
MAJOR ISSUE
The paper needs much more detail description of the sampling procedure and of samples. For now, it is not clear whether the samples were just a loose material scratched off the wall, or monolithic blocks cut off from the wall? How large were the samples? How deep did the sampling penetrate into the wall? How were the samples further pretreated before the analyses – for example, cut into pieces before porosimetry, powdered before TG-TDA, thin sections prepared before microscopy, etc.? All these details must be provided.
- We thank the Reviewer for his / her suggestion, and we have added more information, as requested.
Please also say more about your sampling strategy. For example, why two samples of plaster (LC-1 and LC-2) have been collected instead of one? What was the difference between both sampling spots LC-1 and LC-2 visible in field? Did you try to track some particular difference, e.g. proximity of the damaged zone, intensity of damage etc.? And the same for other types of samples.
For the selection of the samples collected, the Municipality of Comacchio required to collect a few representative samples.
Another issue is that Discussion does only loosely refers to the presented results. Most of the results are not discussed in the Discussion, while them all should be. Please refer to your water content and salt content analyses, and basing on this please present hypotheses on the water circulation (or water and salt circulation) within the walls. Please also refer to the porosimetry analyses and say how the plaster porosity can impact the weathering. And please refer to the TG analyses and say how the composition of plaster may impact the degradation. Please also explain better your conclusion about the degradation of the pigment. Which exactly microscopic observations allow for such conclusion?
- We thank the Reviewer for his / her suggestion, and we have added more information, as requested.
MINOR ISSUES
Page 1, lines 35-43 – it would be worthy to mention also the importance of cultural heritage for tourism and for science (to study the past). And that it has ambiguous economical importance: can provide income serving as a touristic destination, but also requires funds for conservation. Both these economic factors are good reasons to keep the cultural heritage objects in good condition.
We thank the Reviewer’s suggestion and we have added more information, as requested.
Page 4, lines 107-108 – it is not clear how the 2015 campaign could reveal the 2019 damage, that is future in regard to 2015. Please rephrase these sentences.
We agree with the Reviewer, and we have modified the sentence.
Page 4, line 111 (and Figure 3 e,f caption, and anywhere further in the text) – the term “erosion” used here seems not to be the best choice. In general, erosion means the removal of material by erosional agent, such as water flow or wind. Here, it seems that you depict the effect of weathering (in situ deterioration of a rock), which looks at your photos like in situ physical disintegration. This would be a better term.
We thank the Reviewer for his / her suggestion, and we have changed the term in all the manuscript.
Page 4, line 130 – please note that Fig. 4 shows the sampling spots, not kinds of samples.
We thank the Reviewer’s suggestion and we have modified the caption.
Page 5, lines 143-144 – please mark it precisely in Fig. 4 where the conductivity analysis was done.
Figure 4 was modified.
Page 7, lines 214-217 – it is not clear if the information about the water content range in dry, humid etc. masonry is for the Loggiato samples only, or it is a general information about any bricks. If general, then citation(s) is needed. The same for salt content.
For this research work, the water content range in dry, humid, etc. is related to the samples analyzed. For future analysis and observations, we will collect other samples to better understand whether it is a specific problem or extended to the whole structure.
Page 7, lines 241-242 – “revealing that the dosage of the soluble salts was inversely proportional to the humidity” – could you please comment more on this observation in the Discussion section? In general, I would expect different correlations in different water regimes. So, if the water content distribution within the wall was constant over long time, then the zones of low water content are zones of evaporation. And within these zones the concentration of salt in water should be increased locally up to the saturation level. Alternatively, if the water content distribution within the wall was only a temporary situation, caused e.g. by a recent rainfall, then zones of low water content are those that water did not reach (yet) by its capillary action, so the amount of salts should be low due to a short time of dissolution. Can you please refer to such opinion, and also to the possible impact of weather on your observations?
Thanking you for your punctuality, we have considered a number of hypotheses and further investigations will be necessary to investigate the matter further and try to clarify it.
Page 8, line 273 – “uneven distribution of the aggregate (Figs. 7a-b)” - looking at Figure 7 a,b, the distribution of aggregates seems rather even and regular across the field of view.
It was a misunderstanding. The sentence was corrected.
Page 8, lines 273-274 – “The crushed aggregate, with very angular medium sphericity” – there are several quite controversial statements in this sentence. First, what is crushed aggregate? If it was crushed, how do you know that it was originally an aggregate? Or if it is an aggregate right now, how do you know it was crushed? And also, what does it mean “angular sphericity”? Something is either angular or spherical. The same also in lines 284-285.
Also in these cases, it was a misunderstanding. The sentences were corrected.
Page 8, line 276 - “bottom mass is hydraulic in nature” – you probably mean “ground mass” (or “groundmass”). And what does it mean a hydraulic nature? Do you mean the appearance or the parameters? The same also in line 288.
We thank the Reviewer suggestion, and we have corrected the sentences.
Page 9, line 277 – “The adhesion between the matrix and the aggregate” – if you describe the character of contact between the groundmass and larger clasts, please follow the micromorphological terminology provided by Stoops G. 2008. Micromorphology. In: Chesworth W. (Eds)., Encyclopedia of Soil Science. Springer, p. 457-466. The same also in lines 289-290. In your case of LC-1, it looks like gefuric contact (I am not sure about the LC-2 sample, the photograph is not the best).
We thank the Reviewer for his / her suggestions, and we have added more information.
Page 9 line 298 – a typo “bdeposition”
Correct.
Page 9 line 298 – what is the argument behind the blackish staff to be an organic matter, and not the manganous oxide precipitations?
We thank the reviewer for this comment. We tend to think that the black patinas are mainly due to organic material, mainly related to animal feces, as the Loggiato is usually visited by people walking their pets. In order to better define the composition of these black patinas, further and more detailed analyses will certainly be carried out by sampling and SEM-EDS and portable XRF analysis.
Page 10, line 314 – “The plaster samples of the Loggiato dei Cappuccini consisted of different layers” – please show microscope photos (or et least drawings) presenting the spatial relationships between variable layers and their thickness.
In Fig. 7f it is possible to recognize the different layers
Page 10, line 339 – “No powdery blue pigments were observed” – aren’t the leopard spots the pigment powder grains? If not, please provide an explanation or hypothesis what can it be.
We thank the reviewer for this note, surely this aspect needs further investigation in future analytical campaigns
Figure 1 caption – “The Italian map” sounds weird, I would propose “The map of Italy” instead.
Figure 1 caption was corrected.
Figures 5 and 6 – why not to provide color scales for these figures?
We correct the figures adding a color scale.
Figure 6 – it is not clear how these thermograms were constructed. How many measurements exactly were used? Where exactly were these measurements done? This must be specified.
- We indicated in the figures the sampling points.
Figure 7 e,f – why the black patina is not shown at the photographs?
- We correct the figure indicating the black patina
Figure 8 – please indicate where is the external surface.
- We correct the figure indicating the position of the external surface
Table 4 – how could you conclude on the basis of TG analyses that the dolomitic compounds were aggregates? Would other-than-aggregate form of dolomite within a sample produce different TG diagram?
- The classification of dolomitic aggregates is not based only on TG analysis, it’s derived from the correlation between petrographic observation and TG results. We added some lines in the text.
Table 5 – the “Grade of weathering identified” is not explained in the text.
- We thank for the suggestion; we added some more lines for the explanation of the “Grade of weathering identified” in the manuscript
Comments on the Quality of English Language. English is Ok, but long sentences can be divided into shorter ones to improve the readability.
- Following the suggestion of the reviewer, we rearranged the text, trying to make it more readable.
Round 2
Reviewer 3 Report
The revision made the paper much better. There are still some small minor issues to be improved:
Figure 4 caption: “Sampling spot collected” should be replaced by either “Sampling spots” or “Sample collection spots”
Figure 5: in this figure, you show a map of conductivity, which certainly needed at least several points to be measured. While in Fig. 4a, you show only two points (the yellow circles) where the conductivity was measured. Please be consequent. I suggest to show the exact sampling spots in the Figure 5 (just like you did in the Figure 6).
Please also note that two sampling spots shown in Fig. 4a are situated outside of the damaged area, while in Fig. 5 the conductivity is shown only within the damaged area. This needs correction.
Figure 6: Showing the sampling spots is a good improvement of this figure. However, there are two issues that still needs clarification: first, why the colors in the legend are explained as “high damage”, “medium damage” etc., if you measured here the water content and the salt content? The legend should be: “high water content”, “medium water content” etc. And second, why some sampling spots are within the white area? (for example, the upper left one in a), and the medium right one in b) ?) Does it mean that the measurement result was “0”, or that the measurement was not done here?
Figure 7f: Now after marking the position of a black patina the figure is much more readable. Also, some interesting issues arose: the black patina seems to impregnate the whitish layer (of calcite?). And the calcite layer appears to have increased porosity and decreased cohesion toward the patinated zone. So, can you say on the basis of these observations, that you detected a localized chemical weathering of the marble, which is visible as a dissolution of the calcitic material where in contact with black patina? This may have important conclusions for the conservation protocols: cleaning off the patina will weaken the monument, as it will expose the chemically weathered loosened parts of the whitish layer of the rock and thus it will accelerate its disintegration.
The Discussion has been improved and is now satisfactory. Please note that the entire text from lines 346-355 also fits better to the Discussion than to Results.
Line 355: please delete words “surface of the”.
Lines 400-401: the sentence “porous plaster may have a higher salt content due to its ability to absorb and retain moisture from the surrounding environment” needs rephrasing, to avoid mental shortcuts. In fact, the ability to absorb water cannot increase the salt content in the plaster. It may increase the salt content in the absorbed water, due to salts being eluted from the plaster matrix, and thus decrease the salt content in the plaster itself. You can rephrase it for example as “porous plaster may have a higher salt content in the pore water due to its increased ability to absorb and retain moisture from the surrounding environment” or so.
GENERAL: description of the sample preparation methods is still lacking. If you want to avoid making the text longer, you may replace the last column in Table 1, where now are the analytical techniques, by showing here the sample preparation method, for example “thin section preparation for microscopic analyses”, “powdering for TG-DTA” etc.
Author Response
Dear Reviewer, we thank you for your valuable comments.
The revision made the paper much better. There are still some small minor issues to be improved:
Figure 4 caption: “Sampling spot collected” should be replaced by either “Sampling spots” or “Sample collection spots”
We thank the Reviewer for the suggestion, and we have corrected the caption of Figure 4as requested.
Figure 5: in this figure, you show a map of conductivity, which certainly needed at least several points to be measured. While in Fig. 4a, you show only two points (the yellow circles) where the conductivity was measured. Please be consequent. I suggest to show the exact sampling spots in the Figure 5 (just like you did in the Figure 6). Please also note that two sampling spots shown in Fig. 4a are situated outside of the damaged area, while in Fig. 5 the conductivity is shown only within the damaged area. This needs correction.
We are very sorry for the misunderstanding, and we very much thank the Reviewer for the clarification. The two points in Figure 4 are related to the Plaster porosity analysis, and we have added an explanation in the text of paragraph 3.3 and in the caption of Figure 4. Relating Figure 5, we have added sampling spots into the figure as for figure 6.
Figure 6: Showing the sampling spots is a good improvement of this figure. However, there are two issues that still needs clarification: first, why the colors in the legend are explained as “high damage”, “medium damage” etc., if you measured here the water content and the salt content? The legend should be: “high water content”, “medium water content” etc. And second, why some sampling spots are within the white area? (for example, the upper left one in a), and the medium right one in b) ?) Does it mean that the measurement result was “0”, or that the measurement was not done here?
We thank the Reviewer for his / her suggestion, and we have corrected Figure 6 and the respective caption.
Figure 7f: Now after marking the position of a black patina the figure is much more readable. Also, some interesting issues arose: the black patina seems to impregnate the whitish layer (of calcite?). And the calcite layer appears to have increased porosity and decreased cohesion toward the patinated zone. So, can you say on the basis of these observations, that you detected a localized chemical weathering of the marble, which is visible as a dissolution of the calcitic material where in contact with black patina? This may have important conclusions for the conservation protocols: cleaning off the patina will weaken the monument, as it will expose the chemically weathered loosened parts of the whitish layer of the rock and thus it will accelerate its disintegration.
We agreed with the Reviewer, and we widen the discussion on this aspect, both in the results paragraph and in the Discussion paragraph.
The Discussion has been improved and is now satisfactory. Please note that the entire text from lines 346-355 also fits better to the Discussion than to Results.
We agreed with the Reviewer, and we have moved the paragraph into the Discussion section.
Line 355: please delete words “surface of the”.
Deleted.
Lines 400-401: the sentence “porous plaster may have a higher salt content due to its ability to absorb and retain moisture from the surrounding environment” needs rephrasing, to avoid mental shortcuts. In fact, the ability to absorb water cannot increase the salt content in the plaster. It may increase the salt content in the absorbed water, due to salts being eluted from the plaster matrix, and thus decrease the salt content in the plaster itself. You can rephrase it for example as “porous plaster may have a higher salt content in the pore water due to its increased ability to absorb and retain moisture from the surrounding environment” or so.
We very thank the Reviewer for the suggestion, and we have changed the sentence.
GENERAL: description of the sample preparation methods is still lacking. If you want to avoid making the text longer, you may replace the last column in Table 1, where now are the analytical techniques, by showing here the sample preparation method, for example “thin section preparation for microscopic analyses”, “powdering for TG-DTA” etc.
We very much thank the Reviewer for the suggestion, and we have added a new column in Table 1.